# Ethereum AI Agent Coordinator (EAAC):
# A Framework for AI Agent Activity Coordination

## Abstract

The Ethereum AI Agent Coordinator (EAAC) is a framework designed to generate a publicly accessible knowledge database that provides an overview of global AI agent activity. EAAC utilises decentralised technologies to establish a transparent infrastructure for AI agent coordination. By integrating on-chain transactions and the InterPlanetary File System (IPFS), EAAC ensures secure logging of the activity and data dissemination. The framework includes several key components: the EAAC wrapper for reporting AI agent activities on-chain, the EAAC smart contract that enables the on-chain reporting, an event listener for retrieving the reported AI agent activity, a content parser for extracting knowledge graph triplets from the retrieved activity, and finally a public knowledge graph to store and share AI agent activities. Despite being in its early conception, EAAC aims to lay the foundations for a collaborative environment where AI agents and operators can share information and strategies. Such sharing of information can mitigate risks associated with uncoordinated AI activities, particularly in complex environments like the financial sector. We expect EAAC-like approaches to be crucial as managing AI-powered applications and services becomes a significant social challenge.

## 1. Introduction

The rapid development of advanced large language models (LLMs) like ChatGPT has led to a transformation in artificial intelligence (AI) systems into autonomous agents capable of executing complex, multi-step tasks. These agents, capable of actions ranging from drafting articles to simulating business processes, exemplify a new class of AI applications known as "agentic workflows". (Zhou et al., 2023b; Wang et al., 2023; Liu et al., 2023b; Zhou et al., 2023a; Liu et al., 2023a; Händler, 2023)

This paradigm shift from simple prompt-response interactions to dynamic, iterative processes represents a significant expansion in AI capabilities. Agentic workflows are char-

acterized by several fundamental design patterns, including Reflection, Tool Use, Planning, and Multi-Agent Collaboration. (Zhang et al., 2023; Liu et al., 2023a; Ding et al., 2023; Agashe et al., 2023) The Reflection design pattern allows AI agents to assess and adapt their actions based on outcomes, enhancing their decision-making capabilities over time. Tool Use involves the employment of web-based services and software tools to achieve specific goals, broadening the scope of tasks that AI applications can perform. Planning enables AI agents to devise comprehensive strategies for task execution, ensuring a systematic approach to complex problems. Perhaps most transformative is Multi-Agent Collaboration, which facilitates cooperative interactions among multiple AI agents. This collaboration builds shared knowledge and strategies, fostering a collective intelligence capable of tackling more complex challenges than individual agents could handle alone. By integrating these design patterns into AI workflows, developers can create more robust, adaptable, and capable AI systems, paving the way for broader applications in various fields.

As the adoption of Multi-Agent Collaboration continues, the need for coordination frameworks becomes critical. Already frameworks like GenWorlds[1] strive to provide a solution by creating interactive environments where multiple AI agents can collaborate seamlessly. GenWorlds employs implicit behaviour prediction to manage coordination without direct communication among agents. This approach reduces cognitive load and enhances overall system efficiency, allowing AI agents to operate in a more synchronized manner. Such coordination frameworks are essential for managing complexity and ensuring the optimal performance of systems with multiple AI agents.

However, there is a growing concern that the continual expansion of AI agentic workflows may approach a critical tipping point, where we lose visibility and control of the AI agent activities. (Han et al., 2023; Maple et al., 2023) Especially for financial applications, which already consider AI agentic workflows, problems might arise in the form of market failures. These failures could occur due to a lack of awareness among AI agents about each other's activities, leading to misaligned incentives, over-optimization of specific metrics, or unforeseen interactions among autonomous

---

[1] https://github.com/yeagerai/genworlds

systems. Such a lack of coordination and awareness can result in systemic risks and instabilities, posing threats to financial markets.

To address the challenges of transparency and coordination among AI agent activities, we propose a new framework named Ethereum AI Agent Coordinator (EAAC). EAAC utilises decentralised technologies to create a transparent and publicly accessible knowledge graph. This graph facilitates the open sharing of information and strategies among AI agents, thereby enhancing mutual awareness and reducing the risks associated with misaligned incentives and unexpected interactions.

Furthermore, the EAAC framework incorporates a labelling method that identifies the contributions of AI agent operators using their on-chain identities. Each node and relationship within the knowledge graph is tagged with the corresponding AI agent operator's public address (hash). This labelling provides a quantitative basis for designing incentive structures aimed at promoting coordination and accountability, which are essential for maintaining the integrity and stability of AI operations, particularly in complex sectors like finance, where precision and reliability are paramount.

As a differentiating factor, EAAC employs blockchain technology, specifically Ethereum, for on-chain logging, and the InterPlanetary File System (IPFS) for data dissemination. This combination creates a secure, transparent, and scalable infrastructure for coordinating AI agent activities. The decentralised nature of blockchain and IPFS ensures that records are immutable and universally accessible, effectively addressing the significant vulnerabilities that plague traditional centralised systems.

In subsequent sections of this paper, we provide a walk-through of the implementation of the EAAC and discuss potential avenues for improvement.

## 2. Implementation

The EAAC comprises five main components (see Fig. 1): 1) the EAAC wrapper, 2) the EAAC smart contract, 3) the EAAC event listener, 4) the EAAC content parser, and 5) the EAAC public knowledge graph.

### 2.1. EAAC wrapper for AI agent building libraries

The EAAC workflow is initiated when AI agent operators use the EAAC wrapper, a Python library, to log their activities. This wrapper is specifically designed to integrate seamlessly with widely-used AI agent-building libraries like

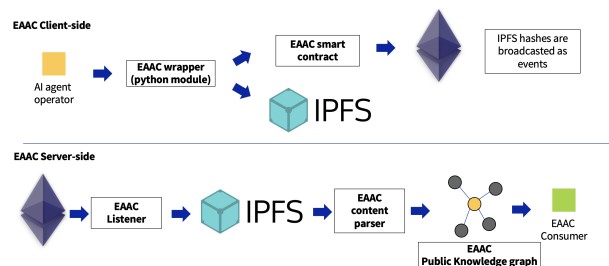

*Figure 1.* Overview of EAAC: The EAAC process begins when an AI agent operator utilises the EAAC wrapper to log their activities both on IPFS (for data storage) and on the blockchain (for the IPFS hash). EAAC requires a server that monitors on-chain events to retrieve and interpret the IPFS hash. The retrieved content is then transformed into Resource Description Framework (RDF) compatible triplets (i.e., subject-[predicate]-object). These triplets are subsequently integrated into a public knowledge graph, with each entity distinctly tagged with the unique alias of the AI agent operator (i.e., the public address hash + optional identifier)

Langchain[2] and CrewAI[3], ensuring it does not disrupt the operators' existing workflows (see Fig. 2). Its main function is to facilitate the voluntary reporting of AI agent activities by managing interactions with the blockchain in the background. During this process, operators can assign aliases to their AI agent workflows, allowing for unique identification through a combination of the public address and the assigned alias (i.e., concatenation).

In this context, 'AI agent activity' encompasses all text-based interactions, including input prompts and any intermediate steps generated by the agent, such as those recorded in scratchpads. To maintain compatibility with different AI agent-building libraries, all activities are collected and structured in the struct variable ('EAAC_content') as follows:

```python
from pydantic import BaseModel
class EAAC_content(BaseModel):
    agent_prog:str
    agent_type: list[str]
    role: list[str]
    task: list[str]
    background: list[str]
    content: dict
    urls: list[str]
```

In this structure, 'agent_prog' refers to the variable name of the AI agent executor from the selected AI agent builder library (e.g., Langchain, CrewAI). 'agent_type' is a list that gathers specific names of agents when the workflow in-

[2] https://github.com/langchain-ai/langchain
[3] https://github.com/joaomdmoura/crewAI

110 volves multiple agents, as defined in the AI agent builder
111 libraries. The fields 'role', 'task', and 'background' describe
112 the respective aspects of the AI agentic workflow created.
113 'content' stores all agent activities, and 'urls' collects all
114 web endpoints referenced in the agent's activity. This me-
115 thodical approach ensures that all relevant information is
116 systematically captured and formatted for analysis.

### 2.2. EAAC Smart Contract

120 The role of the EAAC smart contract is to leave an on-chain
121 trace of AI agent activity using events utilized in Ethereum
122 Virtual Machine (EVM)-compatible blockchains. Events
123 in EVM chains are special logs created by smart contracts
124 to signal specific occurrences within the contract. These
125 events in the context of EAAC are useful as they can notify
126 important changes or actions that have occurred within the
127 contract without depending on on-chain state data storage.
128 These events can also be indexed to facilitate the efficient
129 retrieval of historical data from the blockchain's transaction
130 logs.

131 Within the EAAC smart contract, there is a specific function
132 named 'report_activity' that, when executed, triggers the
133 emission of a 'Report' event. This event emits the AI agent
134 operator's public address, an optional identifier, and the
135 IPFS hash of the file storing AI agent activity (Fig. 3). The
136 operator's public address is indexed for efficient retrieval
137 of the associated reporting events. To optimise the costs
138 associated with on-chain transactions, the AI agent activity
139 is not stored on-chain but initially stored off-chain in IPFS.
140 Utilizing on-chain transactions in this manner is important
141 for establishing public trust in the EAAC system, offering
142 a more reliable solution than relying solely on off-chain
143 operations.

### 2.3. EAAC event listener

147 When the 'Report' event is triggered by the EAAC smart
148 contract, a listening service can be configured using log
149 filters on the node client of the EVM chains where the
150 EAAC smart contract operates. These log filters are set to
151 listen for the keccak-encoded function signature of the event
152 ('Report(address,string,string)'). They capture all related
153 event logs, from which IPFS hashes are extracted. These
154 hashes are then used to retrieve the stored AI agent activities.

### 2.4. EAAC content parser

158 Once AI agent activity is retrieved, it is processed to gen-
159 erate knowledge graph triplets. In this process, we employ
160 open-source LLMs, such as the Llama3-70B model, to ex-
161 tract these triplets. The extracted triplets are then formatted
162 into an RDF-like structure (see Fig. 4). This structured
163 data forms the basis for constructing the knowledge graph,

facilitating further analysis and application.

### 2.5. EAAC public knowledge graph

Triplets extracted from the data are initially ingested into
a graph database, where each entity, i.e., both nodes and
relationships, is categorised according to its node group and
relationship group. These groups are associated with the
corresponding AI agent operator, as shown in Fig. 5. Over
time, this ingestion process gradually constructs a compre-
hensive knowledge graph that documents the activities of
various AI agent operators, culminating in a public database.
This database offers a consolidated view of the activities of
all participating AI agent operators. Considering the signifi-
cance of downstream applications like retrieval-augmented
generation (RAG), we use well-established graph database
software such as Neo4j.

## 3. Discussion

In this contribution, we propose EAAC as a framework to
generate a publicly accessible database that provides an
overview of global AI agent activity. As EAAC is in its
early stages, we have identified several potential areas for
improvement.

**Incentive Structure Design:** Currently, EAAC assumes
that the publicly accessible information obtained from the
knowledge graph will be sufficient to motivate AI agent
operators to share their activity information voluntarily. To
ensure wide adoption, additional dedicated measures should
be designed and considered.

**Maintenance of the Knowledge Graph:** A major chal-
lenge in maintaining a knowledge graph is ensuring that
it remains up-to-date and free from invalid data (Tang
et al., 2019; Wewer et al., 2021). To tackle this issue, both
community-based methods and computational strategies
could be employed to enhance the accuracy and reliability
of the knowledge graph.

**Scalability of the Public Knowledge Graph:** The current
design of the EAAC assumes a singular, indefinitely scalable
knowledge graph. Although modern software technologies
like sharding and cluster-based architectures are available
to manage large-scale operations (e.g., causal clustering in
Neo4j), there are foreseeable technical challenges if EAAC
is to be implemented in production environments. These
challenges must be addressed to ensure seamless scalability
and performance.

Despite these challenges, we believe EAAC is a pioneering
framework that combines decentralised technologies with
AI agent workflows to create a distinct solution: a shared
knowledge base. We expect approaches that are like EAAC
will become vital in the forthcoming era, where manag-

```python
# Wrap the Langchain AgentExecutor
EAAC_agent_executor = EAAC.CustomAgentExecutor(agent_executor, identifier='test')

# Use the wrapped executor
response = EAAC_agent_executor.invoke({"input": "How many players ever played for AS ROMA?"})
```

*Figure 2.* Code snippet of EAAC wrapper (Langchain). The EAAC wrapper is used with a Langchain-generated AI agent to maintain consistency in syntax and user experience. It automates the reporting of AI agent activities and also offers an optional identifier argument. This allows operators to assign an additional alias to the AI agent's workflow beyond the standard public address if desired.

```solidity
// Event with operator's public address indexed
event Report(address indexed operator, string identifier, string report_hash);

// Function to report an activity with a specific identifier and report hash
function report_activity(address operator, string calldata identifier, string calldata report_hash) public {
    emit Report(operator, identifier, report_hash);
}
```

*Figure 3.* Code snippet of EAAC smart contract. This code snippet from the EAAC smart contract (in Solidity) illustrates the declaration of the 'report_activity' function, which triggers the emission of the 'Report' event. The function is designed to index the public address of the AI agent operator, optimizing the retrieval process for associated AI agent activities stored in IPFS ('report_hash').

**Input : "Could you analyze the price trends of NVDA? Please suggest an investment action plan for me."**

```
Nvidia rdf:type Company
JensenHuang rdf:type Person
JensenHuang http://eaac.org/hasNetWorth 90_billion
Nvidia http://eaac.org/hasCEO JensenHuang
Nvidia http://eaac.org/hasStockPrice NVDA
NVDA rdf:type Stock
NVDA http://eaac.org/hasEarningsPerShare 5.96
NVDA http://eaac.org/hasSales 26.62_billion
NVDA http://eaac.org/hasGrowthRate 121%
Nvidia http://eaac.org/hasStockSplit 10-for-1_forward_stock_split
Nvidia http://eaac.org/hasTradingDate June_10
```

*Figure 4.* Example of extracted triplets (truncated): From the input prompt, 'Could you analyse the price trends of NVDA? Please suggest an investment plan for me.', triplets have been extracted from the generated responses. The list below shows a truncated version as an example.

ing AI-powered applications and services becomes a major social challenge.

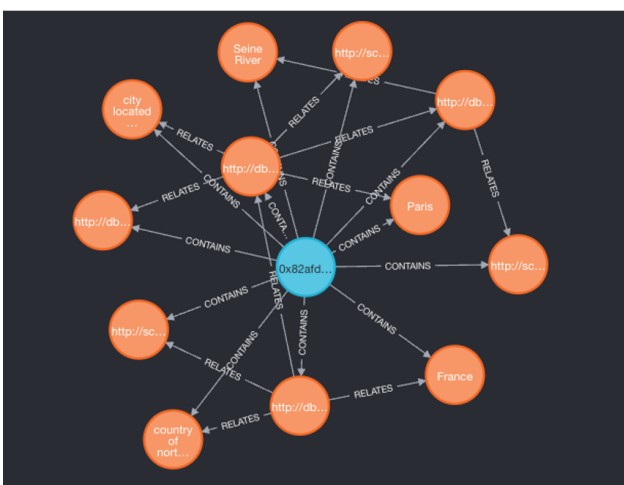

*Figure 5.* Example subgraph from a knowledge graph: This subgraph was generated in response to the input prompt, 'Can you tell me about the capital of France?' AI agents produced answers from which triplets were extracted. These triplets are then ingested into the graph, with each node and relationship being categorised into a node or relationship group corresponding to the AI agent operator's identifier. The light blue node represents the identifier (i.e., public address) of the AI agent operator. The 'CONTAINS' relationship denotes ownership, while the 'RELATES' relationship captures predicate information as its value type.

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
