# OpenReview forum: "Ethereum AI Agent Coordinator (EAAC): A Framework for AI Agent Activity Coordination"
_ICML.cc/2024/Workshop/Agentic_Markets — Agentic Markets @ ICML'24 Poster_

### Official Review · Reviewer_eZvQ · 2024-06-14
**The paper presents the Ethereum AI Agent Coordinator (EAAC), a framework designed to enhance transparency and coordination among AI agents using decentralized technologies. By integrating on-chain transactions and IPFS, it aims to create a publicly accessible knowledge graph of global AI agent activities. The paper aligns with the workshop themes of cooperative AI, Multi-agent Security, Economics, and Finance. It makes a highly original and significant contribution to AI and blockchain integration, with clear and detailed implementation steps, though it could benefit from simplification in certain sections and more standardized references.**

**Rating:** 7
**Confidence:** 3

**Review:**

**Strengths**:
1. Innovative Framework: The integration of blockchain and IPFS for secure and transparent logging of AI agent activities is novel and well-executed. The EAAC framework itself is a unique approach combining blockchain and IPFS for AI agent coordination, applicable across AI, blockchain, and economics domains.
2. Interdisciplinary Applications: The framework addresses a critical and timely topic by tackling uncoordinated AI activities in financial markets.
3. Comprehensive Implementation: The detailed description of the EAAC framework components, including the EAAC wrapper, smart contract, event listener, and content parser, demonstrates a well-thought-out system.
4. Clear and Organized: The paper is clear and well-organized, with a logical flow from the introduction to the implementation details and the discussion of potential improvements. The technical depth and thoroughness in explaining the components and implementation details are commendable.
5. Practical Relevance: The framework addresses real-world challenges in AI coordination, especially in complex environments like the financial sector.
6. Scalability Considerations: The discussion on the scalability and potential challenges of maintaining a public knowledge graph is insightful.

**Areas for Improvement:**
1. Validation: Including preliminary results or a proposed evaluation plan would demonstrate the framework’s effectiveness and robustness.
2. Risk Assessment and Mitigation: Provide more specific strategies for risk assessment and mitigation in agentic markets, and explain how EAAC is able to tackle these issues.
3. Incentive Structures: The paper assumes voluntary participation by AI agent operators, which may not be realistic without strong incentive structures in place. Address this assumption with concrete plans or examples of incentive structures that could encourage broader adoption.
4. Consistent Formatting of References: The references should consistently follow a citation style. Ensure all necessary details (author names, publication year, title, journal name, volume, issue, page numbers) are included for consistency and completeness.

---

### Official Review · Reviewer_yh6G · 2024-06-15
**Review of Ethereum AI Agent Coordination**

**Rating:** 2
**Confidence:** 3

**Review:**

Novel idea for a blockchain coordinated Knowledge Graph indexing AI agents. The problems being addressed are serious and approach seems interesting. However the motivation is not clear, and the paper suffers from a lack of strategic considerations about whether e.g. agents would prefer to reveal and pool information. For instance, the paper notes that there is an open question about whether "publicly accessible information obtained from the knowledge graph will be sufficient to motivate AI agent operators to share their activity information voluntarily. To ensure wide adoption, additional dedicated measures should be designed and considered." However, if the information is publicly available --not for instance, available conditional on sharing their own info--then it's unclear what would be motivating about it.